# Effect of Levetiracetam on Oxidant–Antioxidant Activity during Long-Term Temporal Lobe Epilepsy in Rats

**DOI:** 10.3390/ijms25179313

**Published:** 2024-08-28

**Authors:** Iván Ignacio-Mejía, Itzel Jatziri Contreras-García, Luz Adriana Pichardo-Macías, Mercedes Edna García-Cruz, Blanca Alcira Ramírez Mendiola, Cindy Bandala, Omar Noel Medina-Campos, José Pedraza-Chaverri, Noemí Cárdenas-Rodríguez, Julieta Griselda Mendoza-Torreblanca

**Affiliations:** 1Laboratorio de Medicina Traslacional, Escuela Militar de Graduados de Sanidad, UDEFA, Mexico City 11200, Mexico; ivanignacio402@gmail.com (I.I.-M.); jatziri1984@hotmail.com (I.J.C.-G.); 2Laboratorio de Biología de la Reproducción, Instituto Nacional de Pediatría, Secretaría de Salud, Mexico City 04530, Mexico; 3Departamento de Fisiología, Instituto Politécnico Nacional, Escuela Nacional de Ciencias Biológicas, Mexico City 07738, Mexico; picha_da18@yahoo.com.mx; 4Laboratorio de Neurociencias, Instituto Nacional de Pediatría, Secretaría de Salud, Mexico City 04530, Mexico; ednagcmeg@gmail.com; 5Laboratorio de Farmacología, Instituto Nacional de Pediatría, Secretaría de Salud, Mexico City 04530, Mexico; bramirezmendiola@yahoo.com.mx; 6Laboratorio de Neurociencia Traslacional Enfermedades Crónicas y Emergentes, Escuela Superior de Medicina, Instituto Politécnico Nacional, Mexico City 11410, Mexico; dra.cindy.bandala@gmail.com; 7Departamento de Biología, Facultad de Química, Universidad Nacional Autónoma de México, Mexico City 04510, Mexico; omarnoelmedina@gmail.com (O.N.M.-C.); pedraza@unam.mx (J.P.-C.)

**Keywords:** levetiracetam, temporal lobe epilepsy, oxidative stress, long-term epilepsy, antioxidant enzymes, oxidant markers

## Abstract

Epilepsy is a disorder characterized by a predisposition to generate seizures. Levetiracetam (LEV) is an antiseizure drug that has demonstrated oxidant–antioxidant effects during the early stages of epilepsy in several animal models. However, the effect of LEV on oxidant–antioxidant activity during long-term epilepsy has not been studied. Therefore, the objective of the present study was to determine the effects of LEV on the concentrations of five antioxidant enzymes and on the levels of four oxidant stress markers in the hippocampus of rats with temporal lobe epilepsy at 5.7 months after status epilepticus (SE). The results revealed that superoxide dismutase (SOD) activity was significantly greater in the epileptic group (EPI) than in the control (CTRL), CTRL + LEV and EPI + LEV groups. No significant differences were found among the groups’ oxidant markers. However, the ratios of SOD/hydrogen peroxide (H_2_O_2_), SOD/glutathione peroxidase (GPx) and SOD/GPx + catalase (CAT) were greater in the EPI group than in the CTRL and EPI + LEV groups. Additionally, there was a positive correlation between SOD activity and GPx activity in the EPI + LEV group. LEV-mediated modulation of the antioxidant system appears to be time dependent; at 5.7 months after SE, the role of LEV may be as a stabilizer of the redox state.

## 1. Introduction

Epilepsy, a prevalent neurological disorder affecting 1–2% of the global population, is characterized by a predisposition to generate epileptic seizures [1]. Understanding the underlying mechanisms, particularly in relation to oxidative stress, is crucial for improving therapeutic strategies. Temporal lobe epilepsy (TLE) is the most common type of focal epilepsy in which the medial or internal structures of the temporal lobe are affected. In TLE, spontaneous recurrent seizures (SRSs) usually begin in the hippocampus or a surrounding area affecting one or both temporal lobes of the brain and involve functional and structural changes, such as neuronal loss, axonal sprouting, neurogenesis, synaptogenesis, inflammation, blood–brain barrier leakage and angiogenesis [2,3]. Neuronal hyperexcitability and oxidative injury caused by the excessive production of free radicals may play a role in the initiation and progression of epilepsy [4,5,6]. In humans, a recent study of comparative proteomic profiling of blood plasma revealed the upregulation of proteins related to neuroinflammation and oxidative stress in TLE patients [7]. In another study, bioinformatics analyses of transcriptome databases revealed functional changes in human astrocytes that are specifically related to the response to oxidative stress and inflammation [8].

Diverse antiseizure drugs (ASDs), such as levetiracetam (LEV), brivaracetam and perampanel, are used for the treatment of epilepsy and have recently shown antioxidant properties related to their antiseizure and neuroprotective effects, decreasing lipid oxidation and increasing superoxide dismutase (SOD), glutathione peroxidase (GPx) and catalase (CAT) activities [9,10]. Compared with other ASDs, LEV is a second-generation antiseizure drug that has demonstrated better tolerability and improved efficacy; thus, it has gradually become a first-line drug [11]. The main antiseizure effect of LEV is through synaptic vesicle protein 2A (SV2A). LEV binds to SV2A in both the rat and human brain in a saturable, reversible and stereospecific manner [12]. The mechanism of action of SV2A has not been fully elucidated, but most evidence suggests that SV2A has multiple regulatory effects at various points in the synaptic vesicle cycle, modulating the normal release of neurotransmitters in both inhibitory and excitatory nerve terminals [13,14,15,16,17]. LEV also affects other molecular targets; for example, it blocks voltage-dependent calcium channels [18,19,20,21], reduces potassium currents [22] and reduces calcium transients of ryanodine and inositol trisphosphate (IP_3_) receptors [23,24].

Furthermore, LEV has been shown to have oxidant–antioxidant effects on TLE. In a recent study, LEV administration in rats with TLE significantly increased SOD activity and CAT activity and significantly reduced hydrogen peroxide (H_2_O_2_) levels in comparison with those in epileptic rats without drugs. In addition, LEV showed in vitro scavenging activity against hydroxyl radicals (HO•) [10], and LEV alone significantly increased 8-OHdG and oxidized glutathione (GSSG) levels in the hippocampi of control rats compared with those of epileptic rats [25]. In an in vivo study of the rat hippocampus, microdialysis combined with electron spin resonance spectroscopy revealed that LEV increased ascorbic acid and ∝-tocopherol and decreased inducible nitric oxide synthase (iNOS) [26]. Given the role of oxidative stress in the progression of epilepsy and the demonstrated antioxidant properties of LEV in acute models, there is a critical gap in understanding its effects during long-term epilepsy. Therefore, the objectives of the present study were to determine the effects of LEV on the activities of four antioxidant enzymes (SOD, CAT, GPx, glutathione reductase (GR) and glutathione (GSH) and on the levels of four oxidant stress markers (H_2_O_2_, carbonylated proteins, malondialdehyde (MDA) and 8-hydroxy-2-deoxyguanosine (8-OHdG)) in the hippocampus of a TLE animal model at 5.7 months after status epilepticus (SE). This investigation provides valuable insights into the temporal dynamics of LEV’s antioxidant effects and its potential role in modulating oxidative stress during the chronic stages of epilepsy.

## 2. Results

### 2.1. Antioxidant Markers

The concentrations of SOD, CAT, GPx, GR and GSH antioxidant enzymes were measured in the control (CTRL), CTRL + LEV, epileptic (EPI) and EPI + LEV groups. SOD activity was 54.06 ± 2.85, 46 ± 5.85, 81.5 ± 22.19 and 48.72 ± 6.95 U/mg of protein in the CTRL, CTRL + LEV, EPI and EPI + LEV groups, respectively. SOD activity was significantly greater in the EPI group than in the CTRL (1.50-fold) (*p* < 0.05), CTRL + LEV (1.77-fold) (*p* < 0.01) and EPI + LEV (1.67-fold) (*p* < 0.01) groups, and was notably similar among the CTRL, CTRL + LEV and EPI + LEV groups (Figure 1A).

The measured CAT activities for the CTRL, CTRL + LEV, EPI and EPI + LEV groups were 6.146 ± 3.72, 4.84 ± 1.28, 5.427 ± 1.72 and 4.25 ± 0.94 U/mg protein, respectively. In the CTRL group, the CAT activity was greater than that in the CTRL + LEV (1.26-fold), EPI (1.13-fold) and EPI + LEV groups (1.44-fold). In the EPI group, the CAT activity was slightly greater than that in the CTRL + LEV (1.12-fold) and EPI + LEV (1.27-fold) groups. However, none of these results were significant (Figure 1B).

The GPx activity levels measured in the CTRL, CTRL + LEV, EPI and EPI + LEV groups were 41.79 ± 1.6, 43.33 ± 4.54, 56.55 ± 10.90 and 66.47 ± 6.24 U/mg of protein, respectively. GPx activity was very similar between the CTRL and CTRL + LEV groups. In the EPI and EPI + LEV groups, the activity was similar, but that in the EPI + LEV group was greater than that in the CTRL (1.59-fold) and CTRL + LEV (1.53-fold) groups. However, these differences were not statistically significant (Figure 1C).

The GR concentrations measured for the CTRL, CTRL + LEV, EPI and EPI + LEV groups were 84.98 ± 14.76, 63.05 ± 16.18, 67.60 ± 7.060 and 65.56 ± 9.99 U/mg protein, respectively. The GR activity in the CTRL group was greater than that in the CTRL + LEV (1.34-fold), EPI (1.26) and EPI + LEV (1.30) groups. In addition, the GR activities of the CTRL + LEV, EPI and EPI + LEV groups were very similar, with no significant differences (Figure 1D).

Finally, the GSH levels in the CTRL, CTRL + LEV, EPI and EPI + LEV groups were 2.198 ± 1.291, 3.03 ± 0.53, 2.89 ± 0.92 and 2.58 ± 0.75 µM, respectively. There were no significant differences in the GSH levels among the groups. Additionally, in the CTRL + LEV group, the GSH levels were slightly greater, being 1.38, 1.04 and 1.17 times greater than those in the CTRL, EPI and EPI + LEV groups, respectively (Figure 1E).

### 2.2. Oxidant Markers

Additionally, the concentrations of the stress markers H_2_O_2_, carbonylated proteins, MDA and 8-OHdG were measured. The H_2_O_2_ concentrations were 349 ± 26.02, 584.4 ± 91.13, 484.1 ± 111.7 and 544.8 ± 174.4 ng/mg protein in the CTRL, CTRL + LEV, EPI and EPI + LEV groups, respectively. In the CTRL + LEV group, H_2_O_2_ levels were greater (1.67-fold greater) than those in the CTRL group. In the EPI group, H_2_O_2_ levels were greater (1.39-fold greater) than in the CTRL group. Additionally, H_2_O_2_ levels were slightly greater (by 1.13-fold) in the EPI + LEV group than in the EPI group. However, none of the results were significant (Figure 2A).

The concentrations of the carbonylated protein in the CTRL, CTRL + LEV, EPI and EPI + LEV groups were 10.84 ± 0.72, 11.51 ± 0.35, 12.32 ± 0.41 and 14.23 ± 1.21 nmol/mg protein, respectively. Carbonylated protein levels were greater in the EPI + LEV group than in the CTRL (1.31-fold), CTRL + LEV (1.24-fold) and EPI (1.16-fold) groups. In the EPI group, the levels were greater than those in the CTRL (1.14-fold) and CRTL + LEV (1.07-fold) groups. However, none of the results were significant (Figure 2B).

The MDA concentrations in the CTRL, CTRL + LEV, EPI and EPI + LEV groups were 6.39 ± 0.40, 6.55 ± 0.45, 6.04 ± 0.56 and 6.55 ± 0.31 nmol/mg protein, respectively. Compared with the CTRL (1.03-fold) and EPI (1.06-fold) groups, the CTRL + LEV group presented slightly higher MDA levels; however, the MDA levels in the CTRL + LEV group were similar to those in the EPI + LEV group. However, none of the results were significant (Figure 2C).

Finally, deoxyribonucleic acid (DNA) oxidation was measured by 8-OHdG levels; the values in the CTRL, CTRL + LEV, EPI and EPI + LEV groups were 1.593 ± 0.56, 1.208 ± 0.57, 1.127 ± 0.55 and 1.370 ± 0.92 ng/mg protein, respectively. The 8-OHdG levels were greater in the CTRL group than in the CTRL + LEV (1.32-fold), EPI (1.41-fold) and EPI + LEV (1.16-fold) groups. The levels of 8-OHdG in the EPI + LEV group were greater than those in the CTRL + LEV (1.13-fold) and EPI (1.22-fold) groups. However, these observed changes were not statistically significant (Figure 2D).

### 2.3. Ratio and Correlation Analysis of Biochemical Markers

The biochemical ratios were calculated from the measured parameters. The SOD/H_2_O_2_ ratios in the CTRL, CTRL + LEV, EPI and EPI + LEV groups were 0.165 ± 0.016, 0.080 ± 0.033, 0.198 ± 0.076 and 0.136 ± 0.032, respectively. The SOD/H_2_O_2_ ratio was greater in the EPI group than in the CTRL (1.2-fold), CTRL + LEV (2.48-fold) and EPI + LEV (1.46-fold) groups, with a significant difference in the EPI group compared with the CTRL + LEV group (Figure 3A). The SOD/GPx ratios in the CTRL, CTRL + LEV, EPI and EPI + LEV groups were 1.31 ± 0.171, 1.083 ± 0.136, 1.732 ± 0.900 and 0.739 ± 0.083, respectively; again, the SOD/GPx ratio was greater in the EPI group than in the CTRL (1.32-fold), CTRL + LEV (1.60-fold) and EPI + LEV (2.34-fold) groups, with a significant difference in the EPI group compared with the EPI + LEV group (Figure 3B). In addition, the SOD/(GPx + CAT) ratios were calculated, and the values in the CTRL, CTRL + LEV, EPI and EPI + LEV groups were 1.23 ± 0.155, 0.965 ± 0.084, 1.816 ± 0.677 and 0.704 ± 0.089, respectively. The SOD/(GPx + CAT) ratios were greater in the EPI group than in the CTRL (1.47-fold), CTRL + LEV (1.88-fold) and EPI + LEV (2.58-fold) groups, with a significant difference in the EPI group compared with the other three groups (Figure 3C). Finally, there was a significant positive correlation between SOD activity and GPx activity in the EPI + LEV group (r = 0.970, *p* = 0.001) (Figure 3D).

### 2.4. Levetiracetam Blood Levels

The LEV concentrations in the blood were 22.75 ± 6.3 μg/mL and 22.62 ± 4.9 μg/mL (mean ± SD) in the CTRL + LEV and EPI + LEV groups, respectively. No significant differences were observed between these two groups. These data are consistent with previous reports [10,27].

## 3. Discussion

To study the effects of LEV on antioxidant–oxidant activity in long-term epilepsy, we determined the SOD, CAT, GPx and GR activities; GSH levels; and H_2_O_2_, carbonylated protein, MDA and 8-OHdG oxidant stress marker levels in the hippocampi of TLE model animals at 5.7 months (23 weeks) after SE. The major findings of the present study were as follows: (1) SOD activity was significantly greater in the EPI group than in the CTRL, CTRL + LEV and EPI + LEV groups. (2) No significant differences were found among the oxidant markers among the different groups. However, in the CTRL group, H_2_O_2_ levels and carbonylated proteins were lower than those in the CTRL + LEV, EPI and EPI + LEV groups but greater than those in the 8-OHdG group. (3) Furthermore, the ratios of SOD/H_2_O_2_, SOD/GPx and SOD/GPx + CAT were greater in the EPI group than in the CTRL, CTRL + LEV and EPI + LEV groups, and a positive correlation was observed between SOD and GPx activity in the EPI + LEV group.

In a lithium–pilocarpine-induced TLE animal model, an insult to the brain (caused by SE; the first stage of epilepsy) causes excitotoxicity, neuroinflammation and the production of reactive oxygen species (ROS) and reactive nitrogen species (RNS), which leads to damage to lipids, proteins and DNA; additionally, mitochondrial dysfunction and damage to mitochondrial DNA occur [28,29]. During epileptogenesis (the second stage of epilepsy), several effects, such as lipid peroxidation, hippocampal neurodegeneration, reorganization of neural networks, neuronal hyperexcitability and hypersynchronicity, occur [28,30,31,32]. Notably, during epileptogenesis, very intensive neuronal alterations, including excessive production of ROS/RNS and proinflammatory mediators resulting in increased brain excitability, ultimately lead to neuronal hypersynchronization and increased epileptiform spiking [30]. These factors predispose the brain to SRSs and, consequently, to TLE (the third stage of epilepsy). Although oxidative stress is strongly related to epileptogenesis, few studies have established a link connecting oxidative stress, chronic epilepsy and age [5,33].

In our study, we observed that in long-term TLE, no significant changes in oxidant/antioxidant status occurred, with the exception of SOD, which significantly increased. These findings suggest that in a state where epilepsy is established, the redox state is maintained and does not increase, as observed in the early stages. In this context, the concentration of H_2_O_2_ in the brain increases in the early stages of epilepsy due to the action of SOD, which is responsible for the dismutation of superoxide radicals (O_2_^•−^) into H_2_O_2_, after which the concentration of this oxidant compound decreases due to the action of other antioxidant enzymes [28,34,35,36,37]. Notably, in the past, we suggested H_2_O_2_ as a key oxidant marker of epilepsy because the level of this metabolite significantly increased and was strongly and positively correlated with the number of seizures in epileptic children, probably due to the significant increase in SOD activity also observed in this work [38]. In TLE experimental models, mitochondrial H_2_O_2_ production is also increased [39,40,41]. In addition, in various rat epileptic models, SOD is elevated after 4 h and is maintained for up to 3 weeks [29,39,42,43,44], suggesting that enzyme activity must be maintained over time.

With respect to the oxidant–antioxidant effects of LEV during the progression of epilepsy, LEV was able to maintain lipid peroxidation, nitrite–nitrate and GSH levels and CAT activity at normal values in the hippocampi of animals pretreated with LEV (i.e., before SE) [45,46]. In addition, in another study, pretreatment with LEV decreased the concentrations of MDA compared with those in pilocarpine-untreated animals [47]. Additionally, pretreatment of young pentylenetetrazol-kindled rats with LEV significantly decreased hippocampal MDA and 8-OHdG levels and increased GSH levels and GPx and SOD activities [48]. These results suggest that pretreatment with LEV has a neuroprotective effect that is mediated, at least in part, by the modulation of oxidative stress. During SE, in a self-sustaining SE model, the administration of LEV had a significant positive effect on hippocampal GSH levels compared with those in animals without LEV [49]. In addition, SE induced by lithium–pilocarpine in rats provoked an increase of more than 20-fold in the extracellular concentrations of isoprostanes in the hippocampus, and LEV treatment reduced this oxidative stress marker [50]. During epileptogenesis, LEV significantly decreased nitric oxide and nitric oxide synthase activity and lipid and protein oxidation but significantly increased SOD activity and GSH levels [51]. In another study, LEV did not significantly alter lipid oxidation or GSH levels, but increased nitrite levels [52]. However, in another study, a decrease in GSH levels and an increase in lipid oxidation were observed [53]. Accordingly, inconclusive results on the effects of LEV on epileptogenesis have been reported. In the early stage of epilepsy (14 weeks post-SE), LEV administration significantly increased the activity of SOD and CAT in the hippocampi of epileptic rats compared with that in the hippocampi of untreated epileptic controls, and similar observations have been reported in other studies [54,55]. Moreover, LEV significantly restored GR activity and H_2_O_2_ concentrations compared with those in epileptic rats. In addition, LEV showed in vitro scavenging activity against HO• [10]. These findings suggest that LEV enhances the antioxidant defense system early in the progression of TLE.

In the present study of long-term TLE (23 weeks post-SE), the response to LEV became more complex. Although there was a significant increase in SOD activity in the untreated EPI group compared with the control group (and LEV restored the enzyme values), the administration of LEV did not lead to significant additional changes in other antioxidant enzymes. However, when the relationships between SOD, GPx and CAT activity were analyzed, increased SOD/GPx and SOD/GPx + CAT ratios were observed in the EPI group compared with those in the EPI + LEV and CTRL, CTRL + LEV and EPI + LEV groups, respectively. In addition, we observed a positive correlation between SOD activity and GPx activity in the EPI + LEV group. High SOD/H_2_O_2_ and SOD/GPx ratios indicate high activity and importance of the antioxidant enzyme SOD, likely as a compensatory mechanism to decrease oxidative stress generated by epilepsy, principally through H_2_O_2_, in the chronic stage. These findings agree with previous reports using the pilocarpine SE model, which revealed an increase in CAT, SOD and GPx activities after pilocarpine treatment [56] and an increase in GSH in the hippocampi of adult rats [57]. Moreover, SOD activity also increased in epileptic patients [58]. Chronic oxidative stress in TLE can lead to the upregulation of SOD as the cells attempt to mitigate the damaging effects of O_2_^•−^ [59]. This may indicate an adaptive response of the antioxidant system to chronic oxidative stress in which the system reaches a new equilibrium [60,61].

In particular, the relationship between SOD and H_2_O_2_ had a value less than one, which means that the H_2_O_2_ concentration was greater than the SOD activity. The peroxide produced by SOD is subsequently neutralized by CAT and GPx, resulting in the production of water and oxygen [62,63]. An increased SOD/GPx ratio was detected in the epileptic group, but no significant differences in the SOD/CAT ratio were detected [64]. This finding suggests that epilepsy causes excess H_2_O_2_ production that cannot be neutralized by CAT or GPx. This could cause hippocampal damage in epileptic animals because the accumulated H_2_O_2_ can react with iron (Fenton reaction) to produce highly reactive HO• [62,65]. In contrast, a study in zebrafish assessed the SOD/CAT ratio and revealed that kainic acid induced an imbalance in antioxidant enzyme activity, since an increase in the SOD/CAT ratio was observed in the experimental group compared with the control group [66]. Changes in the equilibrium between H_2_O_2_ formation by O_2_^•−^ dismutation and H_2_O_2_ decomposition by GPx and CAT were evaluated by the SOD/(GPx + CAT) ratio because some studies have suggested that the SOD/GPx + CAT ratio results in better antioxidant ability than either single enzyme alone [67,68]. In our study, a significantly greater SOD/GPx + CAT ratio was detected in the EPI group than in the control group. Although this ratio has not been studied in epilepsy, a study in the brains of aging mice revealed a correlation between an altered SOD/GPx + CAT ratio and increased lipid damage [67,68]. These findings suggest that changes in this ratio could be associated with central nervous system disorders. Similarly, the SOD/(GPx + CAT) ratio is greater in the erythrocytes of Down syndrome patients than in those of controls [69], which may result in increased production of H_2_O_2_. This finding agrees with the SOD/H_2_O_2_ results observed in the epilepsy group. LEV was able to reduce the SOD/H_2_O_2_ ratio and thus H_2_O_2_ levels, probably by maintaining the positive correlation between SOD and GPx activities observed in the EPI + LEV group in long-term epilepsy [10].

Furthermore, we did not observe changes in hippocampal protein carbonylation or lipid peroxidation (measured by MDA) levels among the groups at 14 and 23 weeks of epilepsy progression; however, in a murine model of kainate-induced epilepsy, the carbonylation of mitochondrial proteins increased acutely (48 h) and chronically (6 weeks), coinciding with a decrease in mitochondrial complex I activity [60]. Additionally, previous studies have demonstrated a significant increase in lipid peroxidation during and after SE in animal models [70,71]. This could indicate that protein carbonylation or lipid peroxidation might be more strongly associated with acute phases of oxidative stress or immediate seizure episodes than with the chronic phase of the disease. Similarly, the administration of LEV had limited effects on DNA oxidation (measured by 8-OHdG) in the rat hippocampus at 14 and 23 weeks, indicating that LEV does not have a lasting effect on DNA oxidation in hippocampal homogenates. In contrast, Jarrett et al. (2008) [61] reported sustained increases in 8-OHdG levels in mitochondrial DNA during epileptogenesis, with significant damage observed at 24, 48 and 96 h. Additionally, 8-OHdG levels increased at 21 days and 3 months post-SE, indicating continuous oxidative damage and impaired base excision repair [61,72]. However, they focused exclusively on mitochondrial DNA, whereas our study assessed DNA damage in hippocampal homogenates, which might account for the difference in findings.

In summary, LEV-mediated modulation of the antioxidant system appears to be time-dependent, with significant benefits observed in the early stages of TLE progression. However, its efficacy seems to plateau as the condition becomes chronic, potentially owing to the stabilization of antioxidant mechanisms and the establishment of a new oxidative balance. Chronic conditions likely induce the upregulation of endogenous antioxidant mechanisms as a compensatory response to sustained oxidative damage, and the role of LEV may shift from that of an enhancer to that of a stabilizer of this new redox state [73]. This time-dependent response emphasizes the need for early intervention with LEV in TLE patients, possibly by combining LEV with other therapeutic agents to maintain long-term redox homeostasis [74,75]. Thus, understanding the role of oxidative stress in epilepsy is essential for identifying appropriate therapeutic strategies.

## 4. Materials and Methods

### 4.1. Animals

For this study, male Wistar rats weighing between 250 and 300 g, sourced from Centro de Investigación de estudios avanzados (CINVESTAV), Mexico, were utilized. These animals were housed in a regulated environment with a temperature of 20 ± 2 °C and a light/dark cycle (6:00 a.m./6:00 p.m., corresponding to the light cycle) and were provided with food and water ad libitum. All procedures adhered to the National Institutes of Health (NIH) Guidelines for the Care and Use of Experimental Animals, as well as the Official Mexican Standard of the Secretariat of Agriculture (SAGARPA NOM-062-Z00-1999). The protocols were approved by the Institutional Committee for the Care and Use of Laboratory Animals of the National Institute of Pediatrics (INP 2022/048).

### 4.2. Experimental Design

The rats were randomly assigned to the CTRL, CTRL + LEV, EPI or EPI + LEV groups. Figure 4 describes the experimental design utilized. Briefly, one day before the induction of SE (see details in Section 4.3), all the groups were pretreated with lithium chloride. The next day, the animals were administered methylscopolamine bromide to counteract the peripheral cholinergic effects generated by pilocarpine, and SE was induced with pilocarpine. The rats were subsequently video-monitored to observe SRSs for 9 days. At week 22, osmotic minipumps were implanted to provide subchronic treatment with LEV. Finally, at week 23, a blood sample was taken to measure the concentration of LEV, and the brain was obtained after humane euthanasia to perform assays for antioxidant enzymes and oxidative stress markers.

### 4.3. Animal Model of Temporal Lobe Epilepsy

The procedure for inducing SE and subsequent post-SE care has been documented in earlier publications [27,76]. In summary, the EPI and EPI + LEV groups were pretreated with lithium chloride (127 mg/kg, i.p.; Sigma–Aldrich, St. Louis, MO, USA) 19 h before the administration of pilocarpine. The following day, the animals were given methylscopolamine bromide (1 mg/kg, i.p.; Sigma–Aldrich, St. Louis, MO, USA) half an hour prior to pilocarpine treatment. SE was subsequently triggered via the use of pilocarpine hydrochloride (30 mg/kg, i.p.; Sigma–Aldrich, St. Louis, MO, USA). SE was characterized by sustained convulsive activity lasting more than 5 min. After 90 min of SE, the reaction was stopped with diazepam (5 mg/kg, i.m.; PISA, Mexico City, Mexico), and the animals were immediately placed on a bed of ice for one hour. Eight hours after the initial diazepam dose, a second dose was administered (5 mg/kg, i.m.) along with 5 mL of saline solution (SS; 0.9% s.c.), and the animals were kept overnight at a temperature of 17 °C. The next day, the rats were returned to controlled conditions and were provided with a nutritional feed supplement for 3 days [77,78].

### 4.4. Monitoring of Spontaneous Behavioral Seizures

Twenty-two weeks after the induction of SE, behavioral video-monitoring was conducted on the animals to observe whether the rats exhibited SRSs. For this purpose, the rats were continuously video-monitored (24 h/7 days) via a two-camera system (Steren Mexico, Model Camera CCTV-970, Mexico City, Mexico). Subsequently, well-trained researchers who were blinded to the experimental groups reviewed (by double blind) the videos via the H.264 PlayBack software for Windows (v.1.0.1.15, Infinova, Shenzhen, China) [27,76].

### 4.5. Levetiracetam Treatment

Twenty-two weeks after the induction of SE, for the purpose of providing subchronic treatment with LEV (300 mg/kg/day), ALZET^®^ osmotic minipumps were subcutaneously implanted in the CTRL + LEV and EPI + LEV groups for one week. Briefly, LEV was extracted from a 1000 mg tablet (Pharmalife Laboratories, Atlanta, GA, USA) and dissolved in 3 mL of SS. The solution was then sonicated, centrifuged and filtered prior to use. The osmotic pumps were filled with 2 mL of LEV and incubated for 4 h at 37 °C in SS. The rats were subsequently anesthetized with isoflurane (Sofloran^®^Vet, PISA, Mexico City, Mexico), and the minipumps were then implanted subcutaneously. Finally, an acute dose of LEV (200 mg/kg i.p.; Keppra, UCB Laboratories, Brussels, Belgium) was administered to the rats [77,78,79].

The blood concentration of LEV was measured (n = 4 each group); 7 days after implantation of the osmotic pump, 50 µL of blood was collected from the caudal vein and placed on a Guthrie card (Whatman^®^903, Maidstone, UK). LEV was extracted from the Guthrie card and analyzed by high-performance liquid chromatography according to a previously published method [80].

### 4.6. Tissue Sample Collection and Processing

Twenty-three weeks post SE, the brains were extracted and sectioned to retrieve the hippocampus. The tissues were subsequently frozen and preserved at −80 °C until analysis. Both hippocampi were homogenized in 50 mM phosphate buffer containing 0.05% Triton X-100 (Sigma, St. Louis, MO, USA) at a pH of 7.0 at a 1:5 ratio and then centrifuged at 15,000× *g* for 30 min at 4 °C. The supernatants were isolated and stored in tubes for the assessment of SOD, CAT, GPx, GR, GSH, H_2_O_2_, carbonylated protein, MDA and 8-OHdG levels. On the day of euthanasia, none of the animals exhibited SRSs.

### 4.7. Total Protein Measurement

The protein concentration in the samples was measured using the Lowry technique. In brief, protein levels in these samples were evaluated via an 8-point calibration curve of bovine serum albumin (Sigma, St. Louis, MO, USA), which served as a comparative standard. For analysis, the samples were diluted and added to a microplate, followed by the addition of 250 μL of a solution containing 2% Na_2_CO_3_ (JT Baker, Xalostoc, State of Mexico, Mexico), 0.4% NaOH (JT Baker, Xalostoc, Edo. Mexico, Mexico), 0.02% C_4_H_4_O_6_KNa·4H_2_O (Mallinckrodt, Staines, UK) and 0.01% CuSO_4_ (JT Baker, Xalostoc, Edo. Mexico, Mexico) to each well. This mixture was incubated at ambient temperature for 10 min before the addition of 25 μL per well of 1.0 N Folin & Ciocalteu’s phenol reagent (Sigma, St. Louis, MO, USA). The samples were then promptly mixed and after 30 min, the absorbance was measured at 660 nm [81].

### 4.8. Determination of Antioxidant Markers

#### 4.8.1. SOD Activity Assay

SOD activity was quantified via a superoxide dismutase activity assay kit (Enzo Life Sciences^®^, Butler Pike, Plymouth Meeting, PA, USA) as previously described [38]. Briefly, 25 μL of 1:4 diluted hippocampal supernatant and a blank or SOD standard were added to each well of the assay, along with 150 μL of a Master Mix comprising 10× SOD buffer, WST-1 reagent, xanthine oxidase and distilled water. Additionally, 25 μL of 1× xanthine solution was added to initiate the reaction. The absorbance readings were taken at 450 nm at one-minute intervals for 10 min at room temperature. The results are presented as units per milligram (U/mg) of protein [38].

#### 4.8.2. CAT Activity Assay

The assessment of CAT activity and its calculation were performed following the protocol described by [38], employing a CAT activity assay kit (Enzo Life Sciences^®^, Butler Pike, Plymouth Meeting, PA, USA). Briefly, to initiate the assay in each well, 50 μL of a 1:15 dilution of hippocampal supernatant, a blank or CAT standard, and 50 μL of a 40 μM H_2_O_2_ solution were added. The plate was then incubated for 30 min at room temperature. Subsequently, 100 μL of a reaction mixture composed of detection reagent, 100× HRP and 1× reaction buffer was added to each well. The plate was then further incubated for 10 min. Fluorescence was measured at excitation and emission wavelengths of 530 nm and 590 nm, respectively. The results are expressed in units per milligram (U/mg) of protein [38].

#### 4.8.3. GPx Activity Assay

The determination and measurement of GPx activity were performed in accordance with the methods outlined by [38] via a glutathione peroxidase activity assay kit (Enzo Life Sciences^®^, Butler Pike, Plymouth Meeting, PA, USA). To summarize, for the assay in each well, 20 μL of a 1:4 diluted hippocampal supernatant and either a blank or a GPx standard were added, along with 20 μL of 10× reaction mixture, 140 μL of 1× assay buffer and 20 μL of cumene hydroperoxide to initiate the reaction. The absorbance was recorded at 340 nm at one-minute intervals for 10 min at room temperature. The data are reported as units per milligram (U/mg) of protein [38].

#### 4.8.4. GR Activity Assay

The procedure for measuring GR activity and its calculation was conducted as described previously [38] via a glutathione reductase activity assay kit (Enzo Life Sciences^®^, Butler Pike, Plymouth Meeting, PA, USA). Briefly, for each well of the assay, 50 μL of a 1:5 dilution of hippocampal supernatant and a blank or a GR standard were added, along with 100 μL of a Master Mix consisting of 10× GR buffer, GSSG reagent and distilled water. Additionally, 100 μL of NADP solution was included to initiate the reaction. The absorbance was measured at 340 nm at one-minute intervals over a period of 10 min at room temperature. The results are expressed as units per milligram (U/mg) of protein [38].

#### 4.8.5. GSH Level Determination

The quantification of GSH levels was performed as described by [25] via a glutathione colorimetric detection kit according to the manufacturer’s instructions (Invitrogen™, Thermo Fisher Scientific, Waltham, MA, USA). Briefly, 50 μL of previously prepared 1:8 diluted hippocampal supernatants and a blank or a previously prepared oxidized GSH standard were added to each well. This was followed by the addition of 25 μL of colorimetric detection reagent and 25 μL of reaction mixture to each well. The plate was then incubated at room temperature for 20 min. The absorbance was immediately measured at 405 nm, and the GSH concentration was reported in micromolar (μM) media [25].

### 4.9. Determinatuon of Oxidant Markers

#### 4.9.1. Determination of H_2_O_2_ Levels

The determination of H_2_O_2_ levels was carried out following the protocol of [38] utilizing a hydrogen peroxide colorimetric detection kit (Enzo Life Sciences^®^, Butler Pike, Plymouth Meeting, PA, USA). Briefly, for each assay well, 50 μL of a 1:8 dilution of hippocampal supernatant and a blank or an H_2_O_2_ standard were added, along with 50 μL of sample diluent and 100 μL of color reagent. The plate was then incubated for 30 min at room temperature. The absorbance was recorded at 550 nm. The levels of H_2_O_2_ were reported in nanograms per milligram (ng/mg) of protein [38].

#### 4.9.2. Protein Carbonylation Determination

The measurement of carbonylated protein levels in the samples was conducted using the technique described by [82]. In summary, to remove nucleic acids, the homogenates were treated with 10% streptomycin sulfate for 30 min. Additionally, the homogenates were processed with 10 mM 2,4-dinitrophenyldidrazine (DNPH) (Sigma, St. Louis, MO, USA) and 2.5 M HCl (JT Baker, Xalostoc, Edo. Mexico, Mexico). Following three washes with an ethanol/ethyl acetate mixture (1:1) (JT Baker, Xalostoc, Edo), the protein carbonyl pellet was redissolved in 6 M guanidine hydrochloride (Sigma, St. Louis, MO, USA). The evaluation of carbonyl formation was based on the creation of protein hydrazones through reactions with DNPH. The absorbance was measured at 370 nm. The content of protein carbonyls was quantified as nmol of carbonylated proteins per mg (nmol/mg) of protein [82,83].

#### 4.9.3. MDA Level Determination

The MDA content was quantified via a standard curve of tetramethoxypropane. For the assay, 70 μL of supernatant (Sigma, St. Louis, MO, USA) was combined with 0.25 mL of 10 mM 1-methyl-2-phenylindole (Sigma, St. Louis, MO, USA) and dissolved in an acetonitrile/methanol mixture (3:1) (JT Baker, Xalostoc, Edo. Mexico, Mexico). The reaction commenced upon the addition of 50 μL of 37% HCl (JT Baker, Xalostoc, State of Mexico, Mexico) and the samples were incubated for 40 min at 45 °C before being centrifuged at 3000× *g* for 5 min. The optical density of the resulting supernatant was subsequently measured at 586 nm. The MDA levels are reported as nmol MDA per mg (nmol/mg) of protein [84,85].

#### 4.9.4. 8-OHdG Level Determination

The levels of 8-OHdG were measured using a DNA damage ELISA kit according to the manufacturer’s instructions as described previously [25] (Enzo Life Sciences^®^, Butler Pike, Plymouth Meeting, PA, USA). To summarize the procedure, 50 μL of a 1:8 dilution of hippocampal supernatant and a blank or an 8-OHdG standard, along with 50 μL of anti-8-OHdG, were added to each well. The plate was incubated and washed, followed by the addition of 100 μL of HRP-conjugated anti-mouse IgG to each well. After another round of incubation and washing, 100 μL of tetramethylbenzidine (TMB) substrate was added to each well. The plate was then incubated in the dark, followed by the addition of 100 μL of stop solution to each well. The absorbance was immediately measured at 450 nm. The levels of 8-OHdG are expressed in nanograms per milligram (ng/mg) of protein [25].

### 4.10. Statistical Analysis

The results are displayed as the mean ± standard error of the mean (SEM) for each group of animals. We applied the Shapiro–Wilk normality test to assess whether the distribution conformed to normality. To identify group differences, the data were evaluated via one-way ANOVA with Tukey’s test or the Student–Newman–Keuls test as post hoc tests. Correlation analysis was evaluated by Pearson’s correlation test. All the measurements were conducted in triplicate. A p value of less than 0.05 was considered to indicate statistical significance. Data analysis was performed using SigmaPlot v. 9.5.1 (GraphPad Software Inc., San Diego, CA, USA).

## 5. Conclusions

In conclusion, LEV did not induce significant changes in the activities of the oxidant markers SOD, CAT, GPx and GR or in the levels of the oxidant markers H_2_O_2_, carbonylated proteins, MDA and 8-OHdG in the hippocampi of epileptic rats. However, long-term TLE significantly increased SOD activity and SOD/H_2_O_2_, SOD/GPx and SOD/(CAT + GPx) indices, and LEV administration decreased all these parameters. Furthermore, we observed a significant positive correlation between GPx and SOD activity and LEV administration under epileptic conditions. We suggest that LEV may modulate the antioxidant/oxidant system in a time-dependent manner during the progression of epilepsy, but further studies are needed to elucidate the mechanism of action of LEV in maintaining redox balance during its chronic administration.

## Figures and Tables

**Figure 1 ijms-25-09313-f001:**
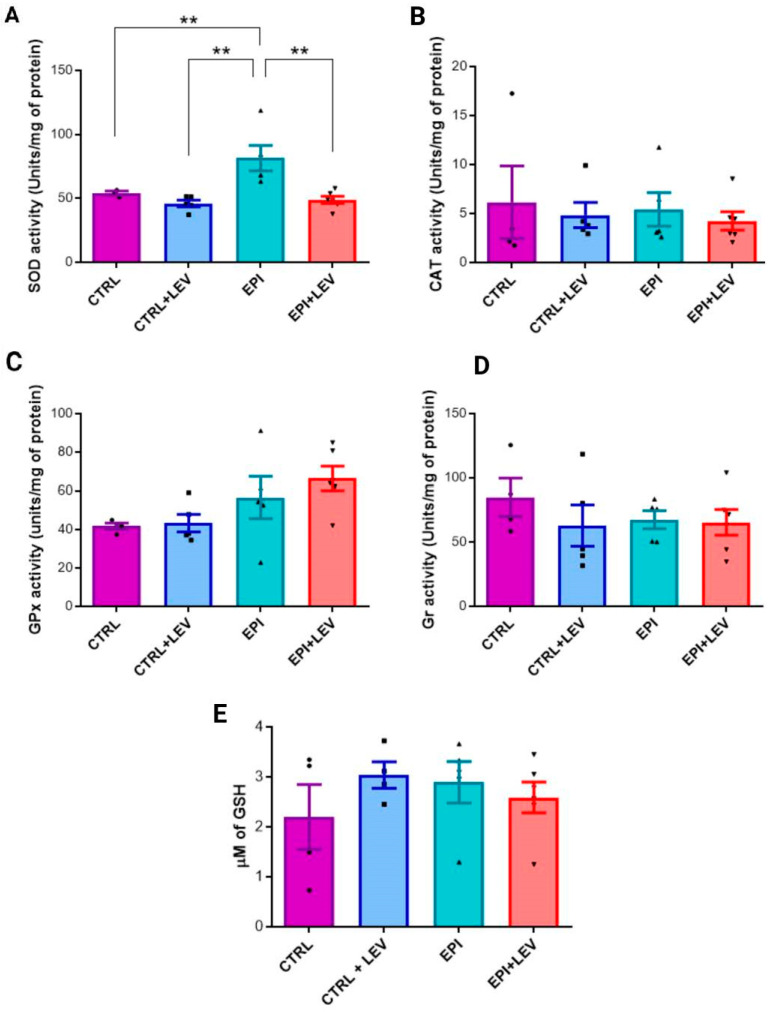
Antioxidant marker levels in the hippocampus of rats. (**A**–**E**) show the values of SOD, CAT, GPx and GR activities and GSH levels, respectively. For all measurements, each quantification was performed in triplicate via data from the CTRL (n = 4), CTRL + LEV (n = 5), EPI (n = 5) and EPI + LEV (n = 6) groups; the values are presented as the means ± SEMs. Differences were analyzed via one-way ANOVA and Tukey’s post hoc test. The brackets indicate the groups that are significantly different. ** *p* < 0.01 indicates a significant difference.

**Figure 2 ijms-25-09313-f002:**
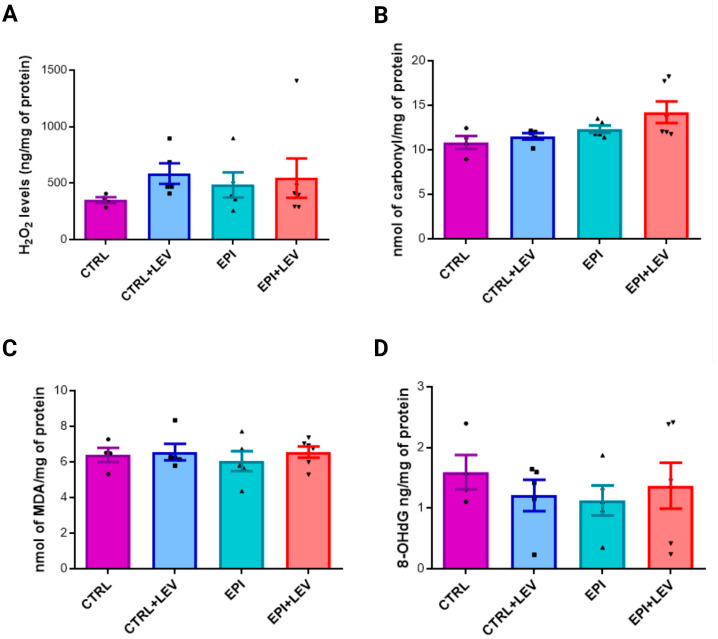
Oxidant levels in the hippocampi of rats. (**A**–**D**) show the oxidant levels of H_2_O_2_, carbonyl proteins, MDA and 8-OHdG, respectively. For all measurements, each quantification was performed in triplicate via data from the CTRL (n = 4), CTRL + LEV (n = 5), EPI (n = 5) and EPI + LEV (n = 6) groups; the values are presented as the means ± SEMs. Differences were analyzed via one-way ANOVA and Tukey’s post hoc test. No significant differences were found among the groups.

**Figure 3 ijms-25-09313-f003:**
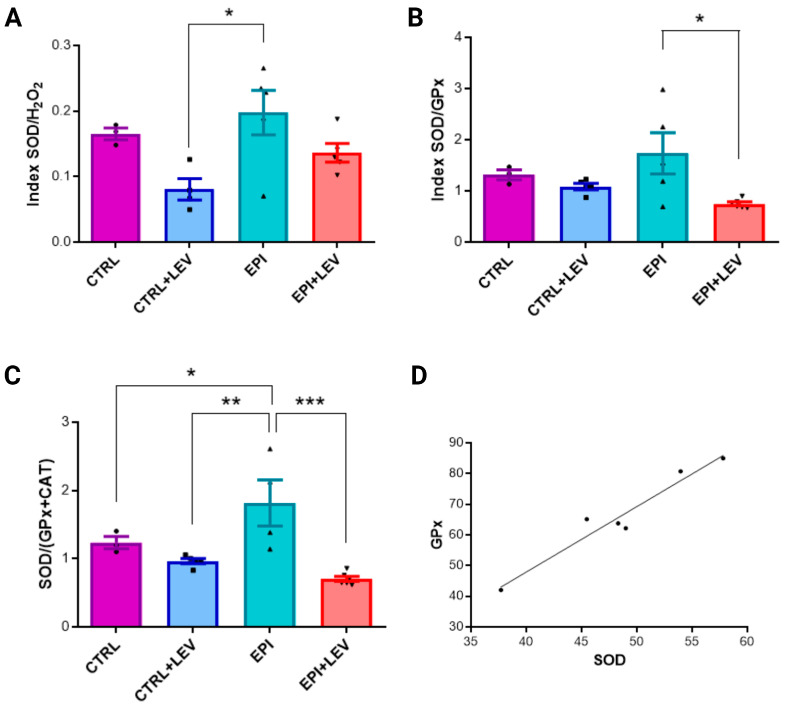
Ratio and correlation analysis of oxidant–antioxidant levels in the hippocampus. (**A**–**C**) show the ratios of SOD/H_2_O_2_, SOD/GPx and SOD (GPx + CAT), respectively. The values are presented as the means ± SEMs. Differences were analyzed via one-way ANOVA and the post hoc Student–Newman–Keuls test. The brackets indicate the groups that are significantly different. * *p* < 0.05, ** *p* < 0.01 and *** *p* < 0.001 indicate significant differences from the CTRL (n = 4), CTRL + LEV (n = 5), EPI (n = 5) and EPI + LEV (n = 6) groups. (**D**) The correlation of GPx/SOD activity in the EPI + LEV group; Pearson’s correlation, *p* < 0.01.

**Figure 4 ijms-25-09313-f004:**
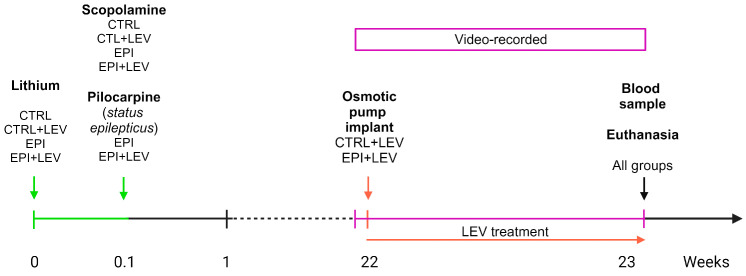
Experimental design. At time 0, all the groups were pretreated with lithium chloride. On day 1, all the animals in the EPI and EPI + LEV groups were administered methylscopolamine bromide, and status epilepticus was induced with pilocarpine. At week 22, the rats were video-recorded for 9 days to observe spontaneous recurrent seizures. Additionally, during this week, osmotic minipumps were implanted in the EPI + LEV and CTRL + LEV groups to provide subchronic treatment with levetiracetam. At week 23, blood and brain samples were obtained. LEV, levetiracetam; CTRL, control; EPI, epileptic.

## Data Availability

The data presented in this study are available upon request from the corresponding author.

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
