# Peer review of "Effect of Levetiracetam on Oxidant–Antioxidant Activity during Long-Term Temporal Lobe Epilepsy in Rats"

_ijms, 2024, doi:10.3390/ijms25179313_

Round 1

Reviewer 1 Report

Comments and Suggestions for Authors

The study aims to investigate the impact of Levetiracetam (LEV) on various antioxidant enzymes and oxidant stress markers in the brains of rats with chronic epilepsy.

The authors found that superoxide dismutase (SOD) activity was significantly higher in the epileptic group (EPI) compared to the control (CTRL), CTRL+LEV, and EPI+LEV groups. No significant differences were found among the oxidant markers. However, the ratios of SOD/hydrogen peroxide (H2O2), SOD/glutathione peroxidase (GPx), and SOD/GPx+catalase (CAT) were more significant in the EPI group than in the CTRL and EPI+LEV groups. There was also a positive correlation between SOD activity and GPx activity in the EPI+LEV group. Additionally, LEV appeared to stabilize the redox state in the epileptic group 5.7 months after SE.

Authors are encouraged to compare enzyme levels based on seizure severity as they possess relevant data, which would enhance the manuscript's quality.

Authors are strongly encouraged to conduct a control experiment by treating rats with scopolamine first, followed by treatment with LEV. 

Also, there is a typo on page 10, line 390.

The text contains some grammatical errors. Please correct them.

I would appreciate it if individual points were shown on the graphs, as it would provide a more detailed view of the data and enhance the clarity of the results. 

The reference to the figure on line 123, page 3, should be corrected to Figure 1B.

I don't believe mice were utilized in this research, but the authors refer to them on line 390, page 10.

It would greatly enhance the clarity of your research if you could include an experimental timeline. This will help readers to better understand and follow your experimental strategy. 

Author Response

Dear Reviewer, 

We carefully considered each comment and made the additional suggested changes to the manuscript. Next, we provide point-by-point responses to each of your requirements:

REVIEWER 1

The authors found that superoxide dismutase (SOD) activity was significantly higher in the epileptic group (EPI) compared to the control (CTRL), CTRL+LEV, and EPI+LEV groups. No significant differences were found among the oxidant markers. However, the ratios of SOD/hydrogen peroxide (H2O2), SOD/glutathione peroxidase (GPx), and SOD/GPx+catalase (CAT) were more significant in the EPI group than in the CTRL and EPI+LEV groups. There was also a positive correlation between SOD activity and GPx activity in the EPI+LEV group. Additionally, LEV appeared to stabilize the redox state in the epileptic group 5.7 months after SE.

  1. Authors are encouraged to compare enzyme levels based on seizure severity as they possess relevant data, which would enhance the manuscript's quality.

R= Thank you for your recommendation. We evaluated the seizure behavior of the rats via the modified Racine scale, which includes 5 levels of seizure intensity: 1) sudden behavioral arrest, motionless staring (with orofacial automatism), 2) head nodding, 3) forelimb clonus with lordotic posture, 4) forelimb clonus, with rearing and falling, and 5) generalized tonic‒clonic activity with loss of postural tone and falling. The results revealed that there were no significant differences among the different levels of seizure intensity and oxidant-antioxidant markers. We also divided the severity of seizures as follows: 1-3 focal, 4-5 generalized. There were also no significant differences in enzyme levels according to the severity of seizures expressed as focal versus generalized.

  1. Authors are strongly encouraged to conduct a control experiment by treating rats with scopolamine first, followed by treatment with LEV.

R= Thank you for your comment: 1) For the generation of status epilepticus, all the groups, including the control group, were administered lithium chloride and scopolamine, and only the EPI and EPI+LEV groups were administered pilocarpine; then, the effect of scopolamine on the generation of oxidative stress was generated in all the groups during the acute phase (we added this information to the materials and methods section 4.2). 2) Rahimzadegan and Soodi (2018) reported that a single dose of scopolamine does not generate an increase in long-term oxidative stress. 3) We obtained our results 5.7 months after status epilepticus was generated (with only one dose of scopolamine used this day). Therefore, our observations are likely due to epilepsy and/or treatment with LEV and not to the effect of scopolamine.

Rahimzadegan, M., & Soodi, M. (2018). Comparison of Memory Impairment and Oxidative Stress Following Single or Repeated Doses Administration of Scopolamine in Rat Hippocampus. Basic and clinical neuroscience, 9(1), 5–14. https://doi.org/10.29252/NIRP.BCN.9.1.5

  1. Also, there is a typo on page 10, line 390.

R= Thank you for your observation; the manuscript has been corrected.

  1. The text contains some grammatical errors. Please correct them.

R= The manuscript has been revised and corrected.

  1. I would appreciate it if individual points were shown on the graphs, as it would provide a more detailed view of the data and enhance the clarity of the results.

R= In accordance with your request, we have changed the graphs.

  1. The reference to the figure on line 123, page 3, should be corrected to Figure 1B.

R= Thank you for your observation; this correction was made (line 108, page 3).

  1. I don't believe mice were utilized in this research, but the authors refer to them on line 390, page 10.

R= This was indeed a mistake and has been corrected.

  1. It would greatly enhance the clarity of your research if you could include an experimental timeline. This will help readers to better understand and follow your experimental strategy.

R= Thank you for your recommendation. We have included the timeline in section 4.2.

Reviewer 2 Report

Comments and Suggestions for Authors

The manuscript investigates the effect of Levetiracetam (LEV) on oxidative stress markers in a rat model of temporal lobe epilepsy (TLE). The study presents intriguing findings regarding the modulation of the antioxidant system by LEV and its potential implications for long-term epilepsy management. However, several key areas require further clarification and improvement to enhance the manuscript's overall quality and scientific rigor.

Major Points

1.       The study reports that SE animals were placed on ice beds and then maintained at 17°C, while control animals did not undergo these procedures. This significant temperature difference could directly affect oxidative stress markers, making comparisons with control animals problematic. Research indicates that hypothermia can alter metabolic and oxidative processes, potentially confounding the results. Please discuss how low temperature might influence oxidative stress markers and why comparable conditions were not applied to control animals. Can you provide a rationale for these methodological choices and discuss relevant literature on the impact of temperature on oxidative stress in epilepsy models?

2.       The manuscript does not provide comprehensive data on seizure parameters such as duration, latency, and frequency, which are crucial for assessing the therapeutic efficacy of LEV. Detailed seizure data are essential for understanding the full impact of LEV treatment, as these parameters are critical for correlating biochemical changes with clinical outcomes. Can you provide detailed seizure data, including duration, latency, and frequency of seizures, observed in the different experimental groups, and discuss how these parameters correlate with the oxidative stress markers measured?

3.       The study utilizes LEV in the form of diluted tablets, which raises concerns about the potential influence of excipients and the appropriateness of this method in an animal model. Typically, research involving animal models employs purified compounds to avoid the interference of excipients present in pharmaceutical formulations designed for human use. The use of diluted tablets may introduce uncontrolled variables that could affect the study's outcomes. What is the rationale behind using diluted LEV tablets instead of a purified compound, and how do the authors ensure that the excipients in the tablets do not interfere with the experimental results? Please discuss the potential impacts of excipients/methodology on the study findings and provide justification for the chosen method.

4.       The methodology section lacks sufficient detail regarding certain experimental procedures, such as the normalization of oxidative stress markers and the specific conditions under which these measurements were taken. Clear and detailed methodological descriptions are vital for reproducibility and for other researchers to validate the findings. The normalization process for oxidative stress markers should be thoroughly explained. Can the authors provide a more detailed description of the normalization process for oxidative stress markers, including the specific controls and standards used? Additionally, explain the rationale behind the chosen methods and how they ensure accurate and reliable results.

5.       The introduction section includes extensive details that seem more appropriate for the discussion section, as it currently blends background information with interpretations and implications that should be reserved for later sections. A well-structured introduction should focus on providing context, stating the research problem, and outlining the study's objectives without delving into detailed interpretations or implications of the results. Please revise the introduction to focus on setting up the research context and objectives, and move detailed discussions and interpretations to the appropriate sections of the manuscript. This will help in maintaining a clear and logical flow of information throughout the paper.

Minor Points

1.       Some claims in the manuscript are not adequately supported by references to current literature, which is essential to back up statements with references to recent and relevant studies. Can the authors include more references to recent studies that support the claims made in the manuscript, particularly regarding the impact of oxidative stress in epilepsy and the use of LEV?

2.       Some conclusions drawn in the manuscript may be overinterpretations of the data presented. It is important to ensure that conclusions are directly supported by the data and to acknowledge any limitations of the study. Could the authors revisit some of the conclusions to ensure they are fully supported by the data, and discuss the limitations that should be considered when interpreting these results?

The manuscript presents an interesting study on the effects of LEV on oxidative stress in a rat model of TLE. To enhance its scientific rigor and clarity, the authors should address the points raised, particularly concerning methodology, use of diluted tablets, comprehensive seizure data, literature support, and data presentation. Ensuring conclusions are directly supported by data will strengthen the manuscript. After addressing these points and providing a point-by-point response to the issues raised, the manuscript can be re-evaluated for potential publication.

Author Response

Dear Reviewer, 

We carefully considered each comment and made the additional suggested changes to the manuscript. Next, we provide point-by-point responses to each of your requirements:

Major Points

  1. The study reports that SE animals were placed on ice beds and then maintained at 17°C, while control animals did not undergo these procedures. This significant temperature difference could directly affect oxidative stress markers, making comparisons with control animals problematic. Research indicates that hypothermia can alter metabolic and oxidative processes, potentially confounding the results. Please discuss how low temperature might influence oxidative stress markers and why comparable conditions were not applied to control animals. Can you provide a rationale for these methodological choices and discuss relevant literature on the impact of temperature on oxidative stress in epilepsy models?

R= Thank you for your comment: 1) Notably, we evaluated oxidative stress in epileptic rats 5.7 months after the onset of status epilepticus and their exposure to low temperatures. 2) In our model, it is essential to keep epileptic animals at low temperatures for one hour in an ice bed and one night at 17 °C because, during status epilepticus, excessive muscle activity generates hyperthermia, and this increase in temperature has been associated with extensive brain damage, increased mortality, and greater difficulty in stopping seizures (Pollandt and Bleck, 2018 and our own experience). The strategy to increase survival and reduce damage to animals is to expose them to low temperatures. Furthermore, if we exposed the control animals to a condition of hypothermia, we would not be equating the conditions with the epileptic rats (since they have hyperthermia), but we would be introducing a variable into the control animals. 3) Djordjevic (2004) suggested that hyperthermia could generate greater oxidative stress than moderate cold stress. The authors evaluated the levels of catalase present in the hippocampus of animals exposed to low (6 °C) or high temperatures (38 °C) for 180 and 60 min, respectively, and observed that catalase activity decreased only when the animal was subjected to high temperatures, whereas when it was kept at 6 °C, the levels of this antioxidant enzyme did not change. Thus, with these arguments, we believe that our observations are due to epilepsy and/or treatment with LEV and not to the effect of one night of low temperature 5 months before the measurement of oxidative stress markers.

  • Pollandt, S., & Bleck, T. P. (2018). Thermoregulation in epilepsy. Handbook of clinical neurology, 157, 737-747.
  • Djordjević, J., Cvijić, G., Vučković, T., & Davidović, V. (2004). Effect of heat and cold exposure on the rat brain monoamine oxidase and antioxidative enzyme activities. Journal of Thermal Biology, 29(7-8), 861-864.
  1. The manuscript does not provide comprehensive data on seizure parameters such as duration, latency, and frequency, which are crucial for assessing the therapeutic efficacy of LEV. Detailed seizure data are essential for understanding the full impact of LEV treatment, as these parameters are critical for correlating biochemical changes with clinical outcomes. Can you provide detailed seizure data, including duration, latency, and frequency of seizures, observed in the different experimental groups, and discuss how these parameters correlate with the oxidative stress markers measured?

R= Thank you for your comment: 1) In our model, which is a chronic condition, it is not possible to measure the latency to the first seizure since LEV is administered months after the induction of status epilepticus. 2) In this work, the therapeutic efficacy of LEV was not evaluated because video recording of rats at least one week before treatment is needed to compare the decrease in the frequency and duration of seizures before and after the administration of the drug. However, owing to technical difficulties, we only have access to videos from two days before and during the treatment with LEV; thus, it was not possible to make this comparison. Nevertheless, with the available data, the frequency and duration of seizures were compared with those of different biochemical markers, and no significant correlations were found. 3) The purpose of this study was to measure the effects of subchronic administration of LEV on different oxidative and antioxidant markers during long-term epilepsy and compare the results with those of the EPI group. Evaluating the effectiveness of LEV under these conditions is a valid and important question, but it would be the subject of another study.

  1. The study utilizes LEV in the form of diluted tablets, which raises concerns about the potential influence of excipients and the appropriateness of this method in an animal model. Typically, research involving animal models employs purified compounds to avoid the interference of excipients present in pharmaceutical formulations designed for human use. The use of diluted tablets may introduce uncontrolled variables that could affect the study's outcomes. What is the rationale behind using diluted LEV tablets instead of a purified compound, and how do the authors ensure that the excipients in the tablets do not interfere with the experimental results? Please discuss the potential impacts of excipients/methodology on the study findings and provide justification for the chosen method.

 R= Thank you very much. We appreciate your comment and your concern regarding the excipients of the pharmaceutical presentation and their possible effects on the animal model. 1) The tablet is the presentation used to treat pediatric patients treated at the National Institute of Pediatrics, and the effects of the excipients (if any) would also be observable in patients. 2) The type and content of excipients in Keppra 500 mg were validated on the website https://www.medicines.org.uk/emc/product/2293/smpc#ref. The composition is described below. The excipients inside the tablet are croscarmellose sodium, macrogol 6000, silica colloidal anhydrous and magnesium stearate, and the excipients of the shell are polyvinyl alcohol. The samples were hydrolyzed with titanium dioxide (E171), macrogol 3350, talc, and iron oxide yellow (E172). These excipients are inert and serve to compress the tablet and stabilize the active ingredient, and they represent only 10% of the net content. We corroborated this in a test where we weighed several tablets prior to pulverization, resulting in an average weight of 1120 mg, of which 1000 mg was the active ingredient (levetiracetam, 89.28%) and 120 mg of the other components described (10.72%). 3) The levetiracetam infusion is prepared in saline solution, in which only hydrophilic molecules are dissolved. The technical data sheets of several excipients indicate that they are insoluble in aqueous solutions: croscarmellose sodium, silica colloidal anhydrous titanium dioxide (E171), talc, and iron oxide yellow (E172). In the last step of preparing the levetiracetam solution, centrifugation is performed to separate these insoluble solids, which are filtered through a 0.45-micron pore to sterilize the solution. This prevents the entry of any insoluble solids and rules out the possibility that several excipients have any effect on the animal model. 4) The economic factor. It was not feasible to acquire the pure chemical substance since for each test, for one animal, 666 mg dissolved in 2 mL of saline solution would be necessary. The price of 120 mg of LEV European Pharmacopeia grade is 189 dollars (1,049 dollars per animal). One hundred milligrams of HPLC-grade LEV cost 447 dollars (2,977 dollars per animal). A total of 200 mg of LEV USP grade cost 650 dollars (2,165 dollars per animal). Additionally, an IV pharmaceutical mixture was proposed; it is available in boxes with 10 ampoules containing 5 mL at a concentration of 100 mg/mL. In addition to the inconvenience of the price ($335 dollars/box), this presentation does not allow the infusion of the dose necessary for our trial since 2 mL (the amount in which the minipump is filled) of Keppra IV contains only 200 mg of active ingredient. 5) The concentration of the saline solution containing LEV was quantified via HPLC. The chromatographic analysis did not detect other molecules, although the method was developed and validated to determine levetiracetam.

  1. The methodology section lacks sufficient detail regarding certain experimental procedures, such as the normalization of oxidative stress markers and the specific conditions under which these measurements were taken. Clear and detailed methodological descriptions are vital for reproducibility and for other researchers to validate the findings. The normalization process for oxidative stress markers should be thoroughly explained. Can the authors provide a more detailed description of the normalization process for oxidative stress markers, including the specific controls and standards used? Additionally, explain the rationale behind the chosen methods and how they ensure accurate and reliable results.

R= In accordance with your request, the methodology for the determination of the different markers has been explained in the corresponding sections and can be replicated under the indicated conditions. In addition, most of the markers were analyzed via a validated kit for each determination (SOD, CAT, GPx, GR, GSH, H2O2 and 8-OHdG). These kits have been used in different models of different scientific studies throughout the world and have reported parameters of sensitivity, reproducibility, recovery, stability and specificity, making them reliable for use. For protein and lipid oxidation determination techniques, no kits have been used, but these techniques have also been reproduced in various studies reported in the literature.

  1. The introduction section includes extensive details that seem more appropriate for the discussion section, as it currently blends background information with interpretations and implications that should be reserved for later sections. A well-structured introduction should focus on providing context, stating the research problem, and outlining the study's objectives without delving into detailed interpretations or implications of the results. Please revise the introduction to focus on setting up the research context and objectives, and move detailed discussions and interpretations to the appropriate sections of the manuscript. This will help in maintaining a clear and logical flow of information throughout the paper.

      R= Thank you for your observation; the introduction has been rewritten.

Minor Points

  1. Some claims in the manuscript are not adequately supported by references to current literature, which is essential to back up statements with references to recent and relevant studies. Can the authors include more references to recent studies that support the claims made in the manuscript, particularly regarding the impact of oxidative stress in epilepsy and the use of LEV?

R= Thank you for your comment. The literature has been updated; however, since some references are original literature, we kept some of these papers plus recent references.

  1. Some conclusions drawn in the manuscript may be overinterpretations of the data presented. It is important to ensure that conclusions are directly supported by the data and to acknowledge any limitations of the study. Could the authors revisit some of the conclusions to ensure they are fully supported by the data, and discuss the limitations that should be considered when interpreting these results?

R= Thank you for your observation; the conclusions have been modified.

  1. The manuscript presents an interesting study on the effects of LEV on oxidative stress in a rat model of TLE. To enhance its scientific rigor and clarity, the authors should address the points raised, particularly concerning methodology, use of diluted tablets, comprehensive seizure data, literature support, and data presentation. Ensuring conclusions are directly supported by data will strengthen the manuscript. After addressing these points and providing a point-by-point response to the issues raised, the manuscript can be re-evaluated for potential publication.

Round 2

Reviewer 2 Report

Comments and Suggestions for Authors

Based on the revised version incorporating the suggested edits and corrections, I find the manuscript suitable for publication.